# Rapid increase in erythropoiesis-stimulating agent resistance is a risk factor for poor renal prognosis in patients with chronic kidney disease pre-dialysis: A BRIGHTEN study sub-analysis

**Shinya Kawamoto**[1]*, **Takao Masaki**[2], **Shoichi Maruyama**[3], **Akane Yamakawa**[4], **Tatsuo Kagimura**[4], **Ichiei Narita**[5]

**1** Department of Cardiovascular Medicine and Nephrology, Dokkyo Medical University Nikko Medical Center, Nikko, Japan, **2** Department of Nephrology, Hiroshima University Hospital, Hiroshima, Japan, **3** Department of Nephrology, Nagoya University, Nagoya, Japan, **4** Translation Research Center for Medical Innovation, Kobe, Japan, **5** Department of Nephrology, Niigata University, Niigata, Japan

* kwmt@dokkyomed.ac.jp

## Abstract

Erythropoiesis-stimulating agent (ESA) resistance is reported in approximately 10% of patients on hemodialysis and is a risk factor for mortality. The BRIGHTEN study conducted on 1,724 non-dialysis patients with chronic kidney disease (CKD) reported an ESA resistance prevalence of 13.3% and was associated with a risk factor for poor cardiovascular and renal prognoses. ESA resistance at 12 weeks after ESA administration can also indicate renal prognoses; however, a single-point estimate is insufficient because ESA resistance increases with CKD progression. We used the BRIGHTEN data for a sub-analysis and found that the ESA resistance index (ERI) (darbepoetin alpha dose/hemoglobin level) transition pattern might be more appropriate for predicting renal prognoses than an initial single-point ERI value. This BRIGHTEN study sub-analysis included 1,625 patients with complete data, comprising 518 dialysis initiations and 18 renal transplants, with a mean follow-up of 1.79 years. The relationship between disease progression and ESA resistance was analyzed using the joint latent class model of longitudinal and time-to-event data. The Kaplan–Meier method was used to estimate the cumulative incidence for the three classes according to the transition pattern of the ERIs. The patient background characteristics in each class were compared. The ERI transition pattern was classified into 3 classes: ERI unchanged, Class A (1,237 patients); moderately increased, Class B (274 patients); and rapidly increased, Class C (114 patients). Class C showed a significantly worse renal survival curve than the other groups, while Class B showed an intermediate survival curve (P<0.0001). The median renal survival times were 1.09 and 0.61 years in Classes B and C, respectively. Patients in Classes B and C were significantly more often male and had diabetic nephropathy, smoking,

**Data availability statement:** All relevant data are within the paper and its Supporting information files.

**Funding:** Translational Research Center for Medical Innovation has received a research grant, which was not specific for this study from Kyowa Kirin Co., Ltd. (KK). Kyowa Kirin was not involved in designing, data interpretation, and manuscript writing for this study, along with any other matters including employment, consultancy, patents, products in development, marketed products, etc. Kyowa Kirin provided support in the form of lecture fees and grants for authors (TM, SM, and IN), but did not have any additional role in the study design, data collection and analysis, decision to publish, or preparation of the manuscript. The specific roles of these authors are articulated in the 'author contributions' section.

**Competing interests:** TM received honoraria, lecture fees, and grants from Kyowa Kirin. SM received honoraria and subsidies or donation from Kyowa Kirin. IN received honoraria and a research grant from Kyowa Kirin as BRIGHTEN STUDY. All the remaining authors declare no conflict of interest. This does not alter our adherence to PLOS ONE policies on sharing data and materials.

hypertension, and more proteinuria than those in Class A. Rapidly increasing ESA resistance patterns predict poor renal prognoses in patients with non-dialysis CKD.

## Introduction

Renal anemia is a common complication in both dialysis and non-dialysis patients with advanced chronic kidney disease (CKD) [1,2]. Erythropoiesis-stimulating agents (ESAs) represent a significant breakthrough in the treatment of renal anemia. ESA resistance is observed in a certain percentage of patients. In the United States, ESA resistance has been reported in 12.5% of 98,972 dialysis patients [3] and is known to be one of the risk factors for mortality [4,5].

In patients with CKD pre-dialysis, the prevalence of ESA resistance is expected to be very close to that in patients on hemodialysis [3]. However, the CKD stages of the cohorts varied, and most of the reports were on the scale of several hundred pre-dialysis patients with CKD [6,7]. Therefore, there are no reports on the exact prevalence among more than 1,000 patients with CKD pre-dialysis.

The BRIGHTEN study (Clinical Trials. gov Identifier: NCT02136563; UMIN Clinical Trial Reg istry Identifier: UMIN000013464.) [8,9], which included 1,724 Japanese patients with CKD pre-dialysis, revealed, for the first time, a large-scale pre-dialysis CKD cohort in which the prevalence of ESA resistance was 13.3%, which was very close to that of previously reported patients on hemodialysis [3]. This study also revealed that initial responsiveness to ESA was statistically significant in predicting renal and cardiovascular disease (CVD) events in patients with CKD pre-dialysis [9].

As CKD progresses over time, hemoglobin (Hb) levels decline despite increasing the dosage of ESAs, indicating ESA hyporesponsiveness. ESA hyporesponsiveness increases with the progression of CKD, reaches a maximum at the time of dialysis initiation, and declines quickly thereafter [10]. After dialysis initiation, ESA hyporesponsiveness remains almost constant; however, in patients with CKD pre-dialysis, ESA hyporesponsiveness increases with the progression of CKD. The BRIGHTEN study [9] revealed that the ESA resistance index (ERI), which is defined as the amount of darbepoietin alpha (DA) (µg) divided by the Hb level (g/dL) 12 weeks after DA administration, was statistically significant in predicting renal and CVD events, with an ERI cut-off value of 5.2. However, the ERI also changed over the long-term follow-up period. Many patients with an initial ERI below 5.2 presented with a renal event during the long-term course of the disease. Although the initial response to ESA can predict renal outcomes to some extent, it may not be sufficient for long-term follow-up. We performed a sub-analysis of the BRIGHTEN study and found that the transition pattern of the ERI might be more appropriate for predicting renal prognoses than an initial single-point ERI.

## Materials and methods

### Study design and patients

The BRIGHTEN study's design, inclusion and exclusion criteria, and other details of the study protocol have been previously described in detail [9]. Briefly, between June

2014 and September 2016, 1,980 patients aged ≥20 years with renal anemia (Hb levels <11 g/dL) and estimated glomerular filtration rates (eGFRs) <60 mL·min$^{-1}$·1.73 m$^{-2}$ were enrolled from 168 institutions. Of these, 1,724 patients were included in the full analysis set of the BRIGHTEN study. In this sub-analysis study, 1,625 patients were analyzed, excluding 37 patients who had observed renal events or mortality events before 12 weeks and 62 patients for whom there were no ERI measurements after 12 weeks (Fig 1).

Renal events were defined as the initiation of maintenance dialysis and kidney transplantation. The BRIGHTEN study included a 50% decrease in the eGFR or an eGFR of <6 mL·min$^{-1}$·1.73 m$^{-2}$, but this sub-analysis included only the initiation of maintenance dialysis and kidney transplantation, which were considered clinical renal deaths.

The ERI was defined as DA dosage (μg) divided by Hb level (g/dL). After the start of DA administration, the ERI was measured over time, and renal events (dialysis initiation/kidney transplantation) were investigated. Patterns of change in the ERI measured over time were spaghetti plotted and analyzed using a joint latent class model (JLCM) for longitudinal and time-to-event data to classify them into three classes [11].

## Statistical analyses

Baseline characteristics are presented as means±standard deviations (SDs), medians (interquartile ranges [IQRs]), or numbers (percentages). We used a JLCM implemented in the "lcmm" R package to estimate the relationship between ERI change trajectories, repeatedly measured after DA administration, and the renal event occurrences.

The JLCM assumes that the population of participants is heterogeneous and consists of homogeneous latent subgroups that share the same trajectory of a repeated measure and the same risk of event occurrence. Using the linear mixed model, we modeled the trajectory of the repeated ERI measurements in this study. The mean trajectory of the repeatedly observed ERIs was estimated using restricted cubic spline curves, which allowed nonlinear changes in the ERI to be considered. The linear mixed model was then combined with the proportional hazard model via the multinomial logistic regression framework, known as the JLCM. In the linear mixed model, we assumed that the parameters were independent of the trajectory classes and did not include covariates other than the measurement points. In order to represent the nonlinear changes in ERI, we used a restricted cubic spline with 4 knots for the linear mixed model.

In the proportional hazard model, we specified "the event" as the renal events defined above and added the age at baseline as a confounder.

We fit the models with one to four classes using a grid search and selected the model with three classes based on the Bayesian Information Criterion and posterior probability of membership.

After fitting the model, the baseline characteristics of the patients in each class were compared. Data are presented as means±SDs, medians (IQRs), or numbers (percentages). Differences between the groups were compared by the analysis of variance for continuous values and the chi-squared test for frequency distributions. Moreover, we plotted the mean trajectory of the observed ERIs to determine whether there was any divergence between the estimated and observed mean trajectories. Additionally, we estimated the cumulative incidence of renal events and performed group comparisons using the log-rank test.

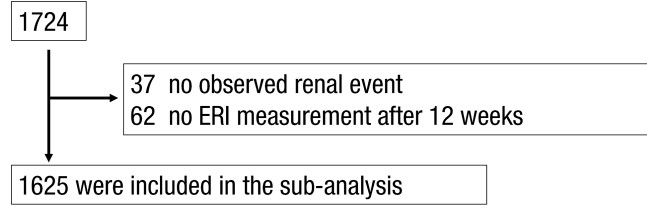

**Fig 1. Flow of participants.**

In addition to classifying latent classes, we performed a multinomial logistic regression analysis to identify the clinical predictors of ERI trajectories. First, we performed multiple imputations on the datasets containing missing measurements and created 100 imputed datasets. We used the mice package in R for multiple imputations. For continuous variables, predictive mean matching was used, while binary variables were imputed using logistic regression. Multinomial logistic models were created for each of the 100 imputed datasets, and variable selections were performed by backward stepwise regression based on the Akaike Information Criteria (AIC) to obtain 100 stepwise results. We selected variables from age, sex, BMI, primary kidney diseases (diabetic nephropathy, nephrosclerosis, and glomerulonephritis), smoking (current, ex-smoker), hypertension, diabetes mellitus, ischemic heart disease, heart failure requiring hospitalization, peripheral vascular disease, baseline usage of drugs (RAS inhibitors, angiotensin II receptor antagonists, ACE inhibitors, and iron supplements), usage of iron supplements at 12 weeks, and clinical and laboratory values at baseline and at 12 weeks (systolic blood pressure, diastolic blood pressure, hemoglobin level, albumin level, serum iron level, ferritin level, transferrin saturation, high-sensitivity C-reactive protein [CRP] level, NT-pro BNP level, and urinary protein creatinine ratio). This resulted in 100 models, each potentially containing different sets of predictors. To identify robust predictors across imputations, we counted the frequency of selection for each predictor across the 100 models. Predictors that were selected more than 70 times in the 100 results were extracted as factors that influenced the classification of the latent classes. To check for multicollinearity, we calculated the correlation coefficients between these predictors in the 100 imputed datasets, checked for the presence of strongly correlated predictor combinations, and selected appropriate predictors. Using only these predictors, we refitted the multinomial logistic model on each of the 100 imputed datasets and pooled the results using Rubin's rule to calculate the adjusted odds ratios for each predictor. The results are summarized in Table 1.

As a supplemental analysis, a similar analysis was performed for each group with baseline eGFRs above and below 15 mL·min$^{-1}$·1.73 m$^{-2}$. Furthermore, as another supplemental analysis, we created a spaghetti plot and cumulative incidence curve of each class for ERIs above and below 5.2, and examined the effect of baseline ERIs on mean trajectories.

All analyses were performed using R 4.1.3 (R Core Team 2022).

## Result

### Patients

The 1,625 patients included in this study were followed up for an average (SD) of 1.79 (0.92) years, and renal events were observed in 518 patients who started dialysis and 18 patients who underwent kidney transplants.

**Table 1. Multinomial logistic regression analysis of ERI trajectory predictors.**

| | Class B | | Class C | |
| --- | --- | --- | --- | --- |
| **Predictor** | **Adjusted OR (95% CI)** | **P-value** | **Adjusted OR (95% CI)** | **P-value** |
| **Sex, male** | 1.61 (1.07–2.42) | 0.022 | 1.59 (0.83–3.04) | 0.16 |
| **Smoking, current smoker** | 0.74 (0.46–1.17) | 0.199 | 1.38 (0.77–2.50) | 0.282 |
| **Ischemic heart disease (%)** | 0.70 (0.45–1.08) | 0.106 | 1.74 (1.02–2.97) | 0.041 |
| **Iron supplementation (baseline)** | 1.91 (1.29–2.83) | 0.001 | 1.24 (0.64–2.38) | 0.520 |
| **Systolic BP at 12 weeks (mmHg)** | 1.00 (0.99–1.01) | 0.700 | 1.01 (1.00–1.03) | 0.029 |
| **Creatinine at 12 weeks (mg/dL)** | 1.38 (1.00–1.90) | 0.050 | 1.40 (0.87–2.25) | 0.161 |
| **Hemoglobin at baseline (g/dL)** | 0.79 (0.66–0.95) | 0.012 | 0.75 (0.57–0.98) | 0.038 |
| **Hemoglobin at 12 weeks (g/dL)** | 0.90 (0.77–1.06) | 0.197 | 0.62 (0.48–0.79) | 0.000 |
| **eGFR at 12 weeks (mL·min$^{-1}$·1.73 m$^{-2}$)** | 0.95 (0.91–1.01) | 0.080 | 0.91 (0.82–1.01) | 0.068 |
| **Urinary protein at 12 weeks (g/gCr)** | 1.11 (1.04–1.18) | 0.001 | 1.13 (1.04–1.22) | 0.004 |

Predictors were selected by backward stepwise variable selection based on the Akaike information criterion. Adjusted ORs were estimated using Class A as the reference group. P-values were calculated using Wald's test.

OR, odds ratio; CI, confidence interval; BP, blood pressure; ERI, erythropoiesis-stimulating agent resistance index; eGFR, estimated glomerular filtration rate.

The ERI transition pattern was divided into 3 classes (Fig 2): Class A (ERI unchanged, 1,237 patients [76.1%]), Class B (ERI moderately increased, 274 patients [16.9%]), and Class C (ERI rapidly increased, 114 patients [7.0%]). Class A (ERI unchanged) had unchanged trajectories at around 4–6 DA dosage/Hb levels throughout the follow-up period. In contrast, Class B (ERI moderately increased) showed a moderate increase from 5 to 20 DA dosage/Hb levels throughout the period. Furthermore, Class C (ERI rapidly increased) exhibited a rapid increase from 7 to 20 DA dosage/Hb levels during the first 50 weeks of follow-up, and after 50 weeks, it increased more rapidly from 20 to 85 DA dosage/Hb levels.

The clinical characteristics of the participants in each class are presented in Table 2. There were significant differences in age, sex, primary kidney disease, smoking rate, the prevalence of hypertension and ischemic heart disease, use of iron preparations, NT-proBNP levels, and urinary protein excretion rates. There were also significant differences in Hb and creatinine (Cr) levels, both at the start of the DA administration and 12 weeks after: Class A, Hb $9.9 \rightarrow 11.1$ g/dL and Cr $2.55 \rightarrow 2.64$ mg/dL; Class B, Hb $9.6 \rightarrow 10.9$ g/dL and Cr $3.57 \rightarrow 3.99$ mg/dL; and Class C, Hb $9.4 \rightarrow 10.3$ g/dL and Cr $3.79 \rightarrow 4.48$ mg/dL.

Classes B and C, with increased ERIs, had a higher proportion of male patients and a greater prevalence of diabetic nephropathy, smoking, and hypertension than Class A.

The predictors of ERI trajectories by multinomial logistic regression analysis are shown in Table 2.

Baseline Hb and urinary protein creatinine ratio (g/gCr) levels at 12 weeks were significant predictors of both Classes B and C ERI trajectories.

It was also suggested that male sex may have had a strong influence on ERI trajectories, particularly for Class B and, to a lesser extent, Class C. It has also been suggested that ischemic heart disease and systolic BP at 12 weeks may have had a strong influence on Class C-like ERI trajectories.

The changes in the observed ERIs in each Class are shown in Fig 3. In this figure, we verify that the estimated mean trajectories of ERIs are not far from the observed mean changes.

The renal survival curves (cumulative incidence curves) for each Class over 4 years are shown in Fig 4.

Class C showed a significantly worse renal survival curve among all three Classes, while Class B showed an intermediate survival curve (P < 0.0001). The median renal survival time was 1.381 and 0.844 years in Classes B and C, respectively.

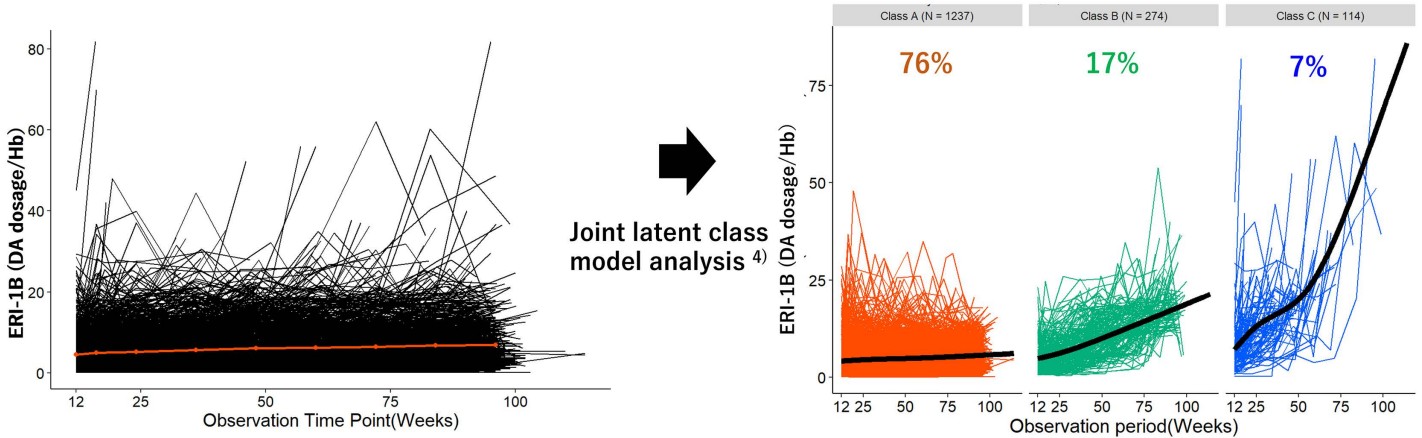

Spaghetti plot of ERI-1B trends for all cases

**Fig 2. ERI transition patterns were divided into three classes.** Class A (ERI unchanged, 1,237 patients [76.1%]), Class B (ERI moderately increased, 274 patients [16.9%]), and Class C (ERI rapidly increased, 114 patients [7.0%]). ERI, erythropoiesis-stimulating agent resistance index.

**Table 2. Baseline clinical characteristics in each class.**

| | Class A | Class B | Class C | P-value |
|---|---|---|---|---|
| n | 1237 | 274 | 114 | |
| Age, years | 70.47 ± 11.75 | 68.43 ± 12.14 | 69.00 ± 12.65 | 0.023 |
| Sex, male (%) | 682 (55.1) | 194 (70.8) | 82 (71.9) | <0.001 |
| BMI, kg/m$^2$ | 22.91 ± 3.93 | 23.95 ± 4.26 | 23.58 ± 4.54 | 0.001* |
| **Primary kidney diseases (%)** | | | | 0.004 |
| Diabetic nephropathy | 314 (25.4) | 89 (32.5) | 47 (41.2) | |
| Nephrosclerosis | 299 (24.2) | 62 (22.6) | 21 (18.4) | |
| Glomerulonephritis | 292 (23.6) | 65 (23.7) | 25 (21.9) | |
| Others | 332 (26.8) | 58 (21.2) | 21 (18.4) | |
| **Smoking (%)** | | | | 0.003 |
| No smoking history | 623 (50.4) | 131 (47.8) | 39 (34.2) | |
| Smoked in the past but not currently | 434 (35.1) | 106 (38.7) | 50 (43.9) | |
| Current smoker | 125 (10.1) | 31 (11.3) | 22 (19.3) | |
| Hypertension (140/90<) (%) | 419 (33.9) | 105 (38.3) | 60 (52.6) | <0.001 |
| Ischemic heart disease (%) | 201 (16.2) | 32 (11.7) | 27 (23.7) | 0.012 |
| Hemoglobin at baseline (g/dL) | 9.88 ± 0.86 | 9.58 ± 0.88 | 9.36 ± 0.90 | <0.001 |
| Hemoglobin at 12 weeks (g/dL) | 11.12 ± 1.11 | 10.89 ± 0.94 | 10.29 ± 1.13 | <0.001 |
| Creatinine at baseline (mg/dL) | 2.55 ± 1.10 | 3.57 ± 1.47 | 3.79 ± 1.20 | <0.001 |
| Creatinine at 12 weeks (mg/dL) | 2.64 ± 1.25 | 3.99 ± 1.66 | 4.48 ± 1.61 | <0.001 |
| Iron at baseline (μg/dL) | 70.87 ± 26.7 | 70.28 ± 25.8 | 70.08 ± 18.7 | 0.914 |
| Transferrin saturation at baseline (%) | 26.60 (10.2) | 27.45 (9.5) | 27.22 (7.1) | 0.398 |
| Ferritin at baseline (ng/mL) | 130.10 ± 131.96 | 143.50 ± 126.87 | 154.20 ± 145.47 | 0.08 |
| High-sensitivity CRP at baseline (ng/mL) | 2620.51 ± 7396 | 2728.00 ± 8851 | 2951.58 ± 10896 | 0.905 |
| Albumin at baseline (g/dL) | 3.77 ± 0.48 | 3.58 ± 0.58 | 3.51 ± 0.51 | <0.001 |
| NT-pro BNP at baseline (pg/mL) | 1156.46 ± 2693.26 | 1517.80 ± 2724.96 | 2166.76 ± 2973.83 | <0.001 |
| Urinary protein creatinine ratio at baseline (g/gCr) | 1.81 ± 2.50 | 3.46 ± 3.39 | 4.36 ± 3.27 | <0.001 |
| ERI ≧ 5.2 at 12 weeks (%) | 331 (26.8) | 95 (34.7) | 65 (57.0) | <0.001 |

Differences between the groups were compared by the analysis of variance for continuous values, the chi-squared test for frequency distributions.

BMI, body mass index; CRP, C-reactive protein; ERI, ERI, erythropoiesis-stimulating agent resistance index.

Significant differences in serum Cr levels were observed between the three Classes at the start (2.6 mg/dL in Class A, 3.6 mg/dL in Class B, and 3.8 mg/dL in Class C). To minimize the effect of baseline serum Cr differences on the renal survival curves, renal survival curves were compared between patients with eGFRs ≥ 15 mL·min$^{-1}$·1.73 m$^{-2}$ and those with eGFRs < 15 mL·min$^{-1}$·1.73 m$^{-2}$. Renal survival curves (cumulative incidence curves) for each latent class by baseline eGFR (≥15 mL·min$^{-1}$·1.73 m$^{-2}$ or <15 mL·min$^{-1}$·1.73 m$^{-2}$) during 4 years are shown in Fig 5. Both eGFRs of ≥15 mL·min$^{-1}$·1.73 m$^{-2}$ and <15 mL·min$^{-1}$·1.73 m$^{-2}$ showed significantly worse renal prognoses in Class C and intermediate prognoses in Class B.

Fig 6 shows a spaghetti plot of each Class divided by 5.2, the cutoff value of the ERI in the first 12 weeks, and their proportions in each class. At 12 weeks, 57%, 34.7%, and 26.8% of the patients had an ERI above the cutoff value of 5.2 in Class C, B, and A, respectively. Class C showed a significantly higher ERI change (above 5.2) at 12 weeks. However, 43% were still below 5.2.

Fig 7 shows the renal survival curves (cumulative incidence curves) for each Class by baseline ERI at 12 weeks (≤5.2 or >5.2) over the 4 years. The overall trend was similar regardless of the baseline ERI, although the proportion of renal

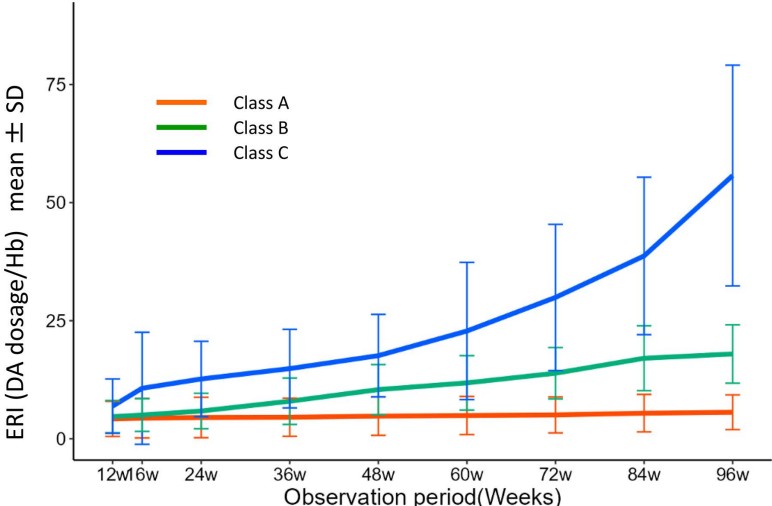

**Fig 3. Changes in ERI in each class.** Class C showed a significantly steeper increase in ERI. ERI, erythropoiesis-stimulating agent resistance index.

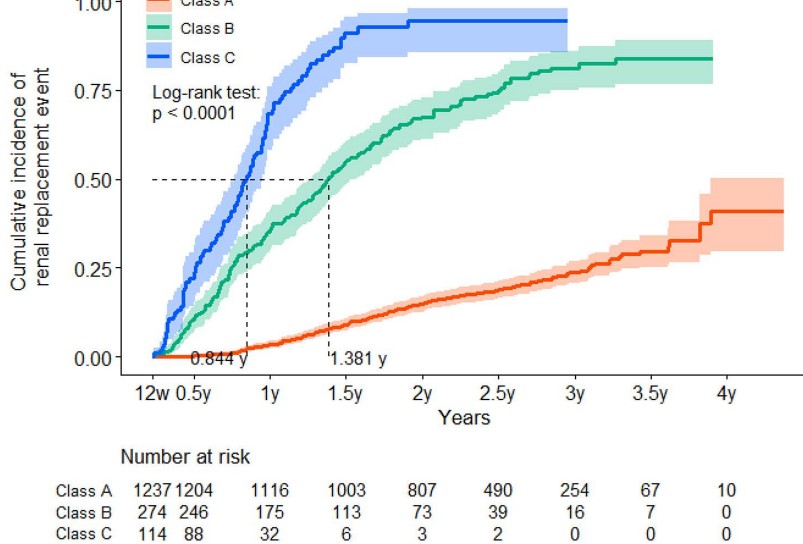

**Fig 4. Renal survival curves (cumulative incidence curves) for each class over 4 years.** Class C showed significantly worse renal survival curves in all 3 groups, and Class B showed intermediate renal survival curves (P < 0.0001). The median renal survival time was 1.381 years in Class B and 0.844 years in Class C.

events was somewhat higher in Class B, with an ERI > 5.2. This suggests that the subsequent transition pattern may have a greater influence on renal prognosis than the baseline ERI.

## Discussion

In patients undergoing hemodialysis, several studies showed that ESA resistance is an independent predictor of mortality and cardiovascular prognosis [12–16]. ESA resistance in patients undergoing hemodialysis remains stable without

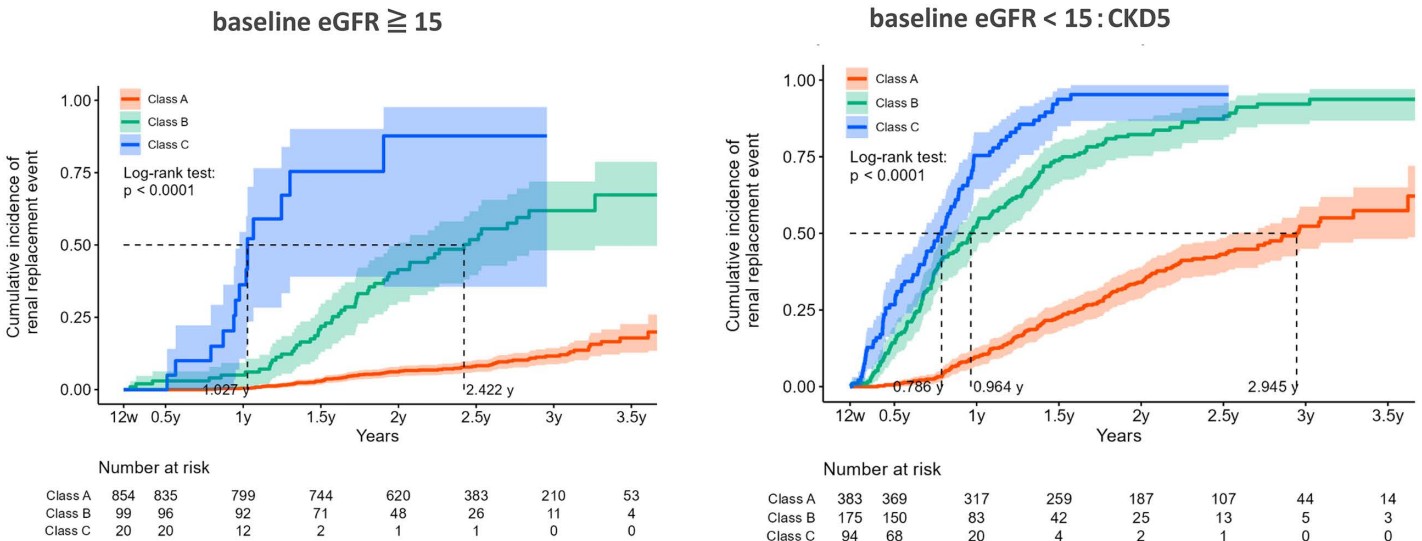

**Fig 5. Both eGFRs ≥ 15 mL·min$^{-1}$·1.73 m$^{-2}$ and <15 mL·min$^{-1}$·1.73 m$^{-2}$ showed significantly worse renal prognoses in Class C, and intermediate renal prognoses in Class B.** eGFR, estimated glomerular filtration rate.

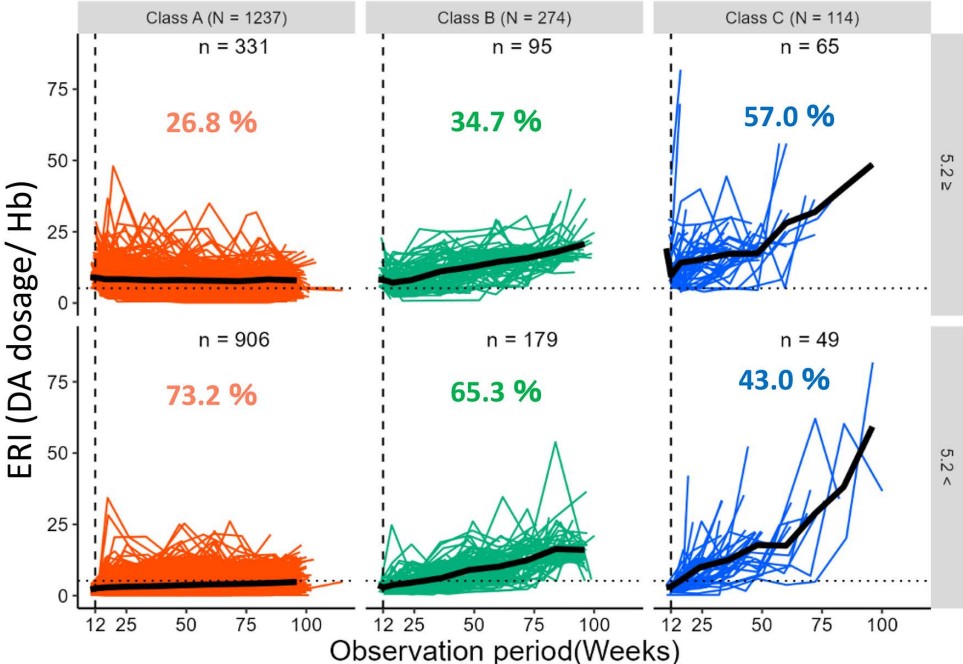

**Fig 6. Class C had a significantly higher percentage of ERIs above 5.2.** ERI, erythropoiesis-stimulating agent resistance index.

significant fluctuations over time, while ESA resistance in patients with CKD who are not undergoing hemodialysis increases with CKD disease progression. Therefore, it is not difficult to predict the prognosis of patients undergoing hemodialysis by cross-sectional evaluation of ESA resistance at a single point. However, it is difficult to predict the prognosis of patients with CKD who are not undergoing hemodialysis, especially in terms of their renal prognosis, based on

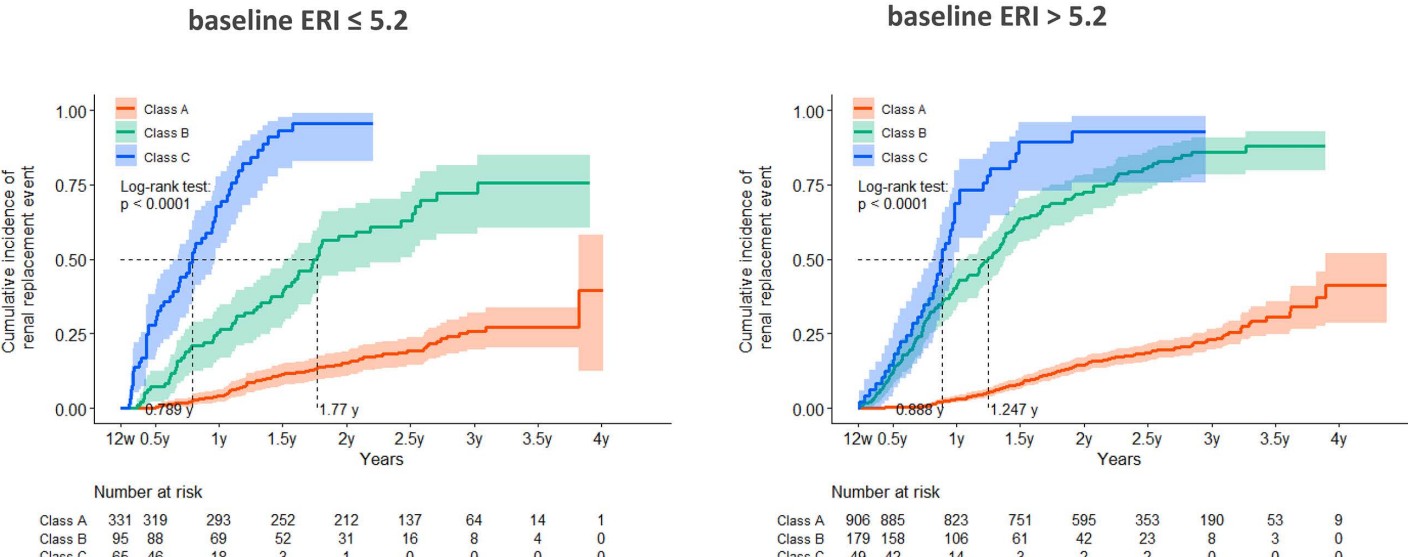

**Fig 7. Renal survival curves (cumulative incidence curves) for each class by baseline ERI at 12 weeks (≤5.2 or >5.2) over 4 years.** The overall trend was similar regardless of baseline ERI. ERI, erythropoiesis-stimulating agent resistance index.

one point of ESA resistance. The BRIGHTEN study evaluated renal and CVD events based on ESA resistance at that time in patients with a mean Hb level of 9.8 g/L and Cr level of 2.6 mg/dL who started to show signs of renal anemia. The BRIGHTEN study showed initial ESA hyporesponsiveness in patients with CKD who were not undergoing hemodialysis associated with renal and CVD events and an optimum cut-off ERI value of 5.2. However, a certain number of patients may progress to end-stage renal disease even with an initial good response (under the cutoff value of the ERI). Tanaka reported that the eGFR negatively correlated with ESA resistance in 513 patients with CKD who were not undergoing hemodialysis [17]. This is consistent with the fact that ESA hyporesponsiveness increases with the progression of CKD, reaching a maximum at the initiation of dialysis and declining rapidly after dialysis initiation [10].

Minutolo et al. reported that 194 patients with CKD who were not undergoing hemodialysis were operationally classified into three groups according to ESA responsiveness and showed that patients in the lowest tertile of responsiveness had poor renal prognoses [18]. The most important factors for hyporesponsiveness are the severity of renal function impairment, iron deficiency, inflammatory state, malnutrition, and hyperparathyroidism [4,19,20]. In addition, other factors, such as diabetes mellitus, proteinuria, and low eGFRs, are associated with lower erythropoietin or Hb levels and may increase the susceptibility of patients to ESA resistance [21–23]. There were significant differences in age, sex, primary kidney disease, smoking rate, prevalence of hypertension and ischemic heart disease, iron preparation use, NT-pro BNP levels, and urinary protein excretion rates among the three classes. Classes B and C showed significantly higher rates of male sex, diabetic nephropathy, smoking, hypertension, and cardiovascular complications, and a greater amount of proteinuria than Class A. Baseline low Hb levels and massive proteinuria at 12 weeks were significant predictors of ERI trajectories for both Classes B and C. Male sex, iron supplementation at baseline, and Cr level at 12 weeks were significant predictors for Class B ERI trajectories. Indicators of arterial stiffness, such as ischemic heart disease and systolic blood pressure at 12 weeks, were significant predictors of Class C ERI trajectories. In Class C, low baseline Hb levels and persistently low Hb levels after ESA administration at 12 weeks were significant predictors.

Not only initial ESA hyporesponsiveness but also a rapid increase in ESA resistance over time is strongly associated with a rapid decline in renal function and correlates with renal outcomes. The ERI at 12 weeks is certainly a convenient

indicator for predicting cardiorenal prognoses; however, the evolution of the ERI during the course of the disease may be a better predictor of renal prognosis. Even if the initial response to ESAs is good, ESA responsiveness deteriorates rapidly with the rapid progression of renal dysfunction in patients with risk factors such as male sex, hypertension, cardiovascular complications, and massive proteinuria. Caution should be exercised in patients with these risk factors for renal dysfunction, as even if the initial response to ESAs is good, the subsequent transition pattern may have a greater impact on renal outcomes than the baseline ERI.

There were also significant differences in the Cr levels at the start of DA administration. We also evaluated the renal survival curve by dividing the patients into two groups with eGFRs above or below 15 mL·min$^{-1}$·1.73 m$^{-2}$ to minimize the effect of baseline Cr levels. Significant differences were observed in the renal survival curves for the three latent classes in both eGFR groups. This means that not only ESA hyporesponsiveness due to poor renal function at the start of renal anemia treatment, but also the transition pattern during the course of the CKD has a significant impact on renal prognosis.

The main limitation of this study was that it was an observational cohort study, and the administration of ESA and iron supplementation was left to the discretion of the physicians in charge of the patient's treatment. Thus, there is a possibility that some physicians may have titrated the ESAs and iron supplements more rapidly than others, and this could have confounded the present observations.

A single spot ERI measured at the start of ESA might be a simpler and potentially useful alternative for predicting renal outcomes. However, while detailed analysis over time, as in this study, can become complex, in actual clinical practice, cases that show rapid development of ESA resistance can be experienced, and in such cases, it is necessary to pay attention to the rapid deterioration in renal function. Therefore, it is meaningful as a large-scale study.

Considering that the ERI passively increases with the deterioration of kidney function, the relationship between the trajectory of ERI and renal outcomes may not function as an independent predictor, but rather could simply reflect the ongoing decline in eGFR.

In this sub-analysis, even if the initial response was good, rapidly increasing ESA resistance during the course of CKD indicated a poor renal prognosis.

## Conclusion

A rapid increase in ESA resistance is a predictor of poor renal prognosis in patients with CKD who are not undergoing hemodialysis.

## Supporting information

**S1 File. CONSORT_2025_editable_checklist (3).**
(DOCX)

**S2 File. Data dictionary of datasets.**
(PDF)

**S3 File. Dataset for JointLCMM.**
(PDF)

**S4 File. Dataset for tables.**
(PDF)

## Author contributions

**Conceptualization:** Shinya Kawamoto, Shoichi Maruyama, Ichiei Narita.

**Data curation:** Akane Yamakawa, Tatsuo Kagimura.

**Formal analysis:** Akane Yamakawa, Tatsuo Kagimura.

**Investigation:** Shinya Kawamoto.

**Methodology:** Shinya Kawamoto, Akane Yamakawa, Tatsuo Kagimura.

**Supervision:** Takao Masaki, Shoichi Maruyama, Tatsuo Kagimura, Ichiei Narita.

**Validation:** Akane Yamakawa, Tatsuo Kagimura.

**Visualization:** Akane Yamakawa.

**Writing – original draft:** Shinya Kawamoto, Akane Yamakawa.

**Writing – review & editing:** Takao Masaki, Shoichi Maruyama, Akane Yamakawa, Tatsuo Kagimura, Ichiei Narita.

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
