## [Decision Letter · Decision Letter 0]

27 Jun 2025

Dear Dr. Kawamoto,

Thank you for submitting your manuscript to PLOS ONE. After careful consideration, we feel that it has merit but does not fully meet PLOS ONE’s publication criteria as it currently stands. Therefore, we invite you to submit a revised version of the manuscript that addresses the points raised during the review process.

We look forward to receiving your revised manuscript.

Kind regards,

Tarek Samy Abdelaziz, MD,FRCP

Academic Editor

PLOS ONE

**Journal Requirements:**

1. When submitting your revision, we need you to address these additional requirements. Please ensure that your manuscript meets PLOS ONE's style requirements, including those for file naming. The PLOS ONE style templates can be found at https://journals.plos.org/plosone/s/file?id=wjVg/PLOSOne_formatting_sample_main_body.pdf and https://journals.plos.org/plosone/s/file?id=ba62/PLOSOne_formatting_sample_title_authors_affiliations.pdf 2. Thank you for stating the following in the Competing Interests section: TM received honoraria, lecture fees, and grants from Kyowa Kirin. SM received honoraria and subsidies or donation from Kyowa Kirin. IN received honoraria and a research grant from Kyowa Kirin. All the remaining authors declare no conflict of interest.  We note that one or more of the authors are employed by a commercial company.  a. Please provide an amended Funding Statement declaring this commercial affiliation, as well as a statement regarding the Role of Funders in your study. If the funding organization did not play a role in the study design, data collection and analysis, decision to publish, or preparation of the manuscript and only provided financial support in the form of authors' salaries and/or research materials, please review your statements relating to the author contributions, and ensure you have specifically and accurately indicated the role(s) that these authors had in your study. You can update author roles in the Author Contributions section of the online submission form. Please also include the following statement within your amended Funding Statement. “The funder provided support in the form of salaries for authors, but did not have any additional role in the study design, data collection and analysis, decision to publish, or preparation of the manuscript. The specific roles of these authors are articulated in the ‘author contributions’ section.”If your commercial affiliation did play a role in your study, please state and explain this role within your updated Funding Statement.  b. Please also provide an updated Competing Interests Statement declaring this commercial affiliation along with any other relevant declarations relating to employment, consultancy, patents, products in development, or marketed products, etc.   Within your Competing Interests Statement, please confirm that this commercial affiliation does not alter your adherence to all PLOS ONE policies on sharing data and materials by including the following statement: "This does not alter our adherence to  PLOS ONE policies on sharing data and materials.” (as detailed online in our guide for authors http://journals.plos.org/plosone/s/competing-interests) . If this adherence statement is not accurate and  there are restrictions on sharing of data and/or materials, please state these. Please note that we cannot proceed with consideration of your article until this information has been declared. Please include both an updated Funding Statement and Competing Interests Statement in your cover letter. We will change the online submission form on your behalf. 3. In the online submission form, you indicated that “The data will be shared on reasonable request to the corresponding author”.  All PLOS journals now require all data underlying the findings described in their manuscript to be freely available to other researchers, either a. In a public repository, b. Within the manuscript itself, or c. Uploaded as supplementary information.This policy applies to all data except where public deposition would breach compliance with the protocol approved by your research ethics board. If your data cannot be made publicly available for ethical or legal reasons (e.g., public availability would compromise patient privacy), please explain your reasons on resubmission and your exemption request will be escalated for approval.

Reviewers' comments:

Reviewer's Responses to Questions

**Comments to the Author**

1. Is the manuscript technically sound, and do the data support the conclusions?

Reviewer #1: Yes

Reviewer #2: Partly

2. Has the statistical analysis been performed appropriately and rigorously?

Reviewer #1: Yes

Reviewer #2: No

3. Have the authors made all data underlying the findings in their manuscript fully available?

Reviewer #1: No

Reviewer #2: Yes

4. Is the manuscript presented in an intelligible fashion and written in standard English?

Reviewer #1: Yes

Reviewer #2: Yes

**Reviewer #1:**  This manuscript reports a secondary data analysis using BRIGHTEN data. I have the following comments and questions.

Lines 150-153, please provide the list of clinical predictors. Please provide a description about the methods used for the imputations of missing measurements as well as how 100 imputed datasets were generated.

Line 163, please provide details about how to pool the results. Do you mean that you put all the selected predictors simultaneously in the model?

Line 169, “Kaplan-Meier curve” should be “cumulative incidence curve”.

Line 217-219, “ischemic heart disease complications” should not be included since it’s increased % is only in class C, not in class B.

Figures 4 and 7 are cumulative incidence curves for renal replacement events. They are not survival curves (Kaplan-Meier curves). Please correct the descriptions in the Result section from lines 236-286.

Lines 276-279, and Figure 7, please correct the inconsistent cutoff of ERI, e.g. >=5.2 or >5.2, <=5.2 or <5.2.

**Reviewer #2: ** This manuscript presents an important sub-analysis of the BRIGHTEN study, examining the association between erythropoiesis-stimulating agent (ESA) resistance trajectory and renal outcomes in pre-dialysis chronic kidney disease (CKD) patients. While the use of a joint latent class model (JLCM) to classify patients by ERI trajectory is novel, several methodological and interpretational concerns limit the strength of the conclusions. Specifically, issues regarding model construction, confounding, outcome definition, and reproducibility require clarification or further analysis.

1. While the study focuses on ERI transition patterns using a longitudinal model, it remains unclear whether this complexity is necessary. A single-point (spot) ERI measured at any time during follow-up might be a simpler and potentially useful alternative for predicting renal outcomes.

2. The exclusion of 37 patients who had no observed renal events is problematic. In time-to-event analyses such as Cox regression or JLCM, patients without events should be retained as censored observations to avoid selection bias.

3. In this sub-analysis, renal events were limited to dialysis initiation and kidney transplantation. However, patients with more advanced kidney dysfunction (i.e., higher ERI) are inherently more likely to experience these endpoints, which may introduce bias. A broader definition of renal events—such as that used in the original BRIGHTEN study, including a ≥50% reduction in eGFR or an eGFR <6 mL/min/1.73 m²—would provide a more sensitive and balanced assessment of disease progression. Inclusion of such endpoints is strongly encouraged to improve generalizability and reduce outcome misclassification.

4. The correct nomenclature is “darbepoetin alfa,” not “darbepoetin” or “darbepoetin α.” The manuscript should be revised for accurate drug names.

5. The worse prognosis in Class C may not be due to ERI trajectory itself, but rather to confounding by inflammation, malnutrition, or iron deficiency—common features in this group. The current analysis does not separate these effects.

6. The classification into Classes A–C is described qualitatively (“stable,” “moderately increasing,” “rapidly increasing”) without quantitative criteria such as slope, rate of change, or spline coefficients. This hinders reproducibility.

7. Given that ERI increases passively with worsening renal function, the association between ERI trajectory and renal outcomes may merely reflect ongoing eGFR decline rather than serve as an independent predictor.

8. Class C had significantly lower baseline eGFR than other classes. Since ERI and eGFR are inversely correlated, the observed ERI trajectory may act as a proxy for renal function. Subgroup analysis for eGFR <15 is insufficient; continuous eGFR should be included as a covariate or time-dependent variable in the model.

**Do you want your identity to be public for this peer review?** For information about this choice, including consent withdrawal, please see our Privacy Policy

Reviewer #1: No

Reviewer #2: No

---

## [Author Response · Author response to Decision Letter 1]

30 Sep 2025

Reviewer #1: This manuscript reports a secondary data analysis using BRIGHTEN data. I have the following comments and questions.

Lines 150-153, please provide the list of clinical predictors. Please provide a description about the methods used for the imputations of missing measurements as well as how 100 imputed datasets were generated.

Thank you for your important comments. We have added the list of clinical predictors (p.11, lines 136-144). Furthermore, we have added information to the manuscript (p.10, lines 144-147) regarding how we performed multiple imputation to handle missing data using the mice package in R, specifically employing the mice.par function for parallel processing. The imputation process involved generating 100 imputed datasets (m = 100) to ensure adequate reflection of the uncertainty due to missingness and to improve the stability and accuracy of subsequent analyses.

The imputation was run with a maximum of 15 iterations (maxit = 15) to allow the algorithm to converge. For continuous variables, predictive mean matching (PMM) was used, while binary variables were imputed using logistic regression (logreg). The seed was set to 1234 to ensure reproducibility.

Line 163, please provide details about how to pool the results. Do you mean that you put all the selected predictors simultaneously in the model?

Thank you for your essential feedback. In this study, multiple imputation was performed using the mice package in R to handle missing data, generating 100 imputed datasets. Continuous variables were imputed using predictive mean matching, while binary variables were imputed via logistic regression.

For each imputed dataset, a multinomial logistic regression model was fitted with variable selection conducted by backward stepwise regression based on the Akaike Information Criterion (AIC). This resulted in 100 models, each potentially containing different sets of predictors.

To identify robust predictors across imputations, we counted the frequency of selection for each predictor across the 100 models. Predictors selected in at least 70% of the models (i.e., in 70 or more datasets) were considered stable and included in the final set of variables.

Subsequently, multinomial logistic regression models were refitted on each of the 100 imputed datasets using only these stable predictors simultaneously.

This approach ensures that variable selection reflects the uncertainty due to missing data, and the final inference is based on models including predictors consistently selected across imputations.

Line 169, “Kaplan-Meier curve” should be “cumulative incidence curve”.

Thank you for your comment. We have corrected it throughout the manuscript (p.11, line 157 & p.21, 238,).

Line 217-219, “ischemic heart disease complications” should not be included since it’s increased % is only in class C, not in class B.

Thank you for your comment. We have rewritten this statement (p.19, lines 204-206

).

Figures 4 and 7 are cumulative incidence curves for renal replacement events. They are not survival curves (Kaplan-Meier curves). Please correct the descriptions in the Result section from lines 236-286.

Thank you for your comment. We have rewritten this section as well (p.20, line 223,226, p.22 line 258,264).

Lines 276-279, and Figure 7, please correct the inconsistent cutoff of ERI, e.g. >=5.2 or >5.2, <=5.2 or <5.2.

Thank you for your comment. We have corrected this statement (p.24, lines 291-300).

Reviewer #2: This manuscript presents an important sub-analysis of the BRIGHTEN study, examining the association between erythropoiesis-stimulating agent (ESA) resistance trajectory and renal outcomes in pre-dialysis chronic kidney disease (CKD) patients. While the use of a joint latent class model (JLCM) to classify patients by ERI trajectory is novel, several methodological and interpretational concerns limit the strength of the conclusions. Specifically, issues regarding model construction, confounding, outcome definition, and reproducibility require clarification or further analysis.

1. While the study focuses on ERI transition patterns using a longitudinal model, it remains unclear whether this complexity is necessary. A single-point (spot) ERI measured at any time during follow-up might be a simpler and potentially useful alternative for predicting renal outcomes.

Certainly, while a detailed analysis, such as the one in this study, may seem complex, it can be perceived in actual clinical practice that cases showing rapid ESA resistance should be noted for rapid progression of renal dysfunction, and I believe it would be beneficial as a large-scale longitudinal detailed study would be meaningful. We added the following text to the “Discussion” section.

A single spot ERI measured at the start of ESA might be a simpler and potentially useful alternative for predicting renal outcomes. However , while detailed analysis over time, as in this study, can become complex, in actual clinical practice, cases that show rapid development of ESA resistance can be experienced, and in such cases, it is necessary to pay attention to the rapid deterioration in renal function. Therefore, it is meaningful as a large-scale study. (p 26, lines 333–338 in the Discussion section)

2. The exclusion of 37 patients who had no observed renal events is problematic. In time-to-event analyses such as Cox regression or JLCM, patients without events should be retained as censored observations to avoid selection bias.

We agree with your assessment. We have revised the text (p. 7, lines 87) because our text was insufficient.

We excluded patients who had observed renal events or death events before 12 weeks and patients for whom there were no ERI measurements after 12 weeks, because we were interested in the progression of ERI after 12 weeks and its correlation with events.

3. In this sub-analysis, renal events were limited to dialysis initiation and kidney transplantation. However, patients with more advanced kidney dysfunction (i.e., higher ERI) are inherently more likely to experience these endpoints, which may introduce bias. A broader definition of renal events—such as that used in the original BRIGHTEN study, including a ≥50% reduction in eGFR or an eGFR <6 mL/min/1.73 m²—would provide a more sensitive and balanced assessment of disease progression. Inclusion of such endpoints is strongly encouraged to improve generalizability and reduce outcome misclassification.

It is true that significant differences were observed in the baseline Cr values of each class, and since this study was limited to serious renal events such as renal replacement events, a more sensitive and balanced assessment of disease progression would be to include a ≥50% reduction in eGFR or an eGFR <6 mL/min/1.73 m² in order to describe poor renal prognosis in the strict sense. We recognize that a study limited to serious renal events is insufficient, but we have interpreted this as meaning that those who did not experience a renal replacement event over a 4 year observation period did not demonstrate poor prognosis.

4. The correct nomenclature is “darbepoetin alfa,” not “darbepoetin” or “darbepoetin α.” The manuscript should be revised for accurate drug names.

Thank you for your comment. We have corrected this phrase (p.6, lines 70).

5. The worse prognosis in Class C may not be due to ERI trajectory itself, but rather to confounding by inflammation, malnutrition, or iron deficiency—common features in this group. The current analysis does not separate these effects.。

There are significant differences in CRP, Ferritin, and Alb levels, which may be involved in the poor response to ESA and may be confounding factors. However, since these are not listed in Table 1 as predictors of rapid deterioration in renal function, this suggests that the increase in ERI accompanying the deterioration of renal function mentioned in #7 may not solely be passive, and we cannot completely deny the influence of these factors.

6. The classification into Classes A–C is described qualitatively (“stable,” “moderately increasing,” “rapidly increasing”) without quantitative criteria such as slope, rate of change, or spline coefficients. This hinders reproducibility.

We totally agree with your assessment. We have clarified the specific ERI values within the trajectories of Classes A-C (p.14, lines 177-183).

7. Given that ERI increases passively with worsening renal function, the association between ERI trajectory and renal outcomes may merely reflect ongoing eGFR decline rather than serve as an independent predictor.

We cannot deny the possibility that this is the case. We have stated this in the Discussion section in the text below.

Considering that ERI passively increases with the deterioration of kidney function, the relationship between the trajectory of ERI and renal outcomes may not function as an independent predictor, but rather could simply reflect the ongoing decline in eGFR. �pp. 27, lines 339-342

8. Class C had significantly lower baseline eGFR than other classes. Since ERI and eGFR are inversely correlated, the observed ERI trajectory may act as a proxy for renal function. Subgroup analysis for eGFR <15 is insufficient; continuous eGFR should be included as a covariate or time-dependent variable in the model.

Although not included in the paper, similar classification results and trajectories were obtained in a model that included eGFR as a time-dependent covariate in the common fixed effect of the linear mixed model. The figures and table below show the results using the model with the eGFR included.

---

## [Decision Letter · Decision Letter 1]

23 Oct 2025

Rapid increase in erythropoiesis-stimulating agent resistance is a risk factor for poor renal prognosis in patients with chronic kidney disease pre-dialysis: A BRIGHTEN study sub-analysis

PONE-D-25-20232R1

Dear Dr. Kawamoto,

We’re pleased to inform you that your manuscript has been judged scientifically suitable for publication and will be formally accepted for publication once it meets all outstanding technical requirements.

Kind regards,

Tarek Samy Abdelaziz, MD,FRCP

Academic Editor

PLOS ONE

Additional Editor Comments (optional):

Reviewers' comments:

Reviewer's Responses to Questions

**Comments to the Author**

Reviewer #1: All comments have been addressed

2. Is the manuscript technically sound, and do the data support the conclusions?

Reviewer #1: (No Response)

3. Has the statistical analysis been performed appropriately and rigorously?

Reviewer #1: (No Response)

4. Have the authors made all data underlying the findings in their manuscript fully available?

Reviewer #1: (No Response)

5. Is the manuscript presented in an intelligible fashion and written in standard English?

Reviewer #1: (No Response)

Reviewer #1: (No Response)

**Do you want your identity to be public for this peer review?** For information about this choice, including consent withdrawal, please see our Privacy Policy

Reviewer #1: No

---

## [Editor Report · Acceptance letter]

PONE-D-25-20232R1

PLOS ONE

Dear Dr. Kawamoto,

I'm pleased to inform you that your manuscript has been deemed suitable for publication in PLOS ONE. Congratulations! Your manuscript is now being handed over to our production team.

Kind regards,

on behalf of

Professor Tarek Samy Abdelaziz

Academic Editor

PLOS ONE